# Occupational Factors and Socioeconomic Differences in Breast Cancer Risk and Stage at Diagnosis in Swiss Working Women

**DOI:** 10.3390/cancers14153713

**Published:** 2022-07-29

**Authors:** Jean-Luc Bulliard, Nicolas Bovio, Patrick Arveux, Yvan Bergeron, Arnaud Chiolero, Evelyne Fournier, Simon Germann, Isabelle Konzelmann, Manuela Maspoli, Elisabetta Rapiti, Irina Guseva Canu

**Affiliations:** 1Center for Primary Care and Public Health (Unisanté), University of Lausanne, 1010 Lausanne, Switzerland; nicolas.bovio@unisante.ch (N.B.); patrick.arveux@unisante.ch (P.A.); simon.germann@unisante.ch (S.G.); irina.guseva-canu@unisante.ch (I.G.C.); 2Neuchâtel and Jura Cancer Registry, 2000 Neuchâtel, Switzerland; manuela.maspoli@ne.ch; 3Fribourg Cancer Registry, 1701 Fribourg, Switzerland; yvan.bergeron@liguessante-fr.ch; 4Population Health Laboratory (#PopHealthLab), University of Fribourg, 1700 Fribourg, Switzerland; arnaud.chiolero@ovs.ch; 5Valais Cancer Registry, Valais Health Observatory, 1950 Sion, Switzerland; isabelle.konzelmann@ovs.ch; 6Geneva Cancer Registry, University of Geneva, 1211 Geneva, Switzerland; evelyne.fournier@unige.ch (E.F.); Elisabetta.Rapiti@unige.ch (E.R.)

**Keywords:** breast cancer, occupation, socioeconomic status, risk, incidence, stage, inequalities, Switzerland

## Abstract

**Simple Summary:**

Socioeconomic status and occupation affect the risk of breast cancer, the most common cancer in women, but whether the effect is the consequence of work exposure or of socioeconomic status is often difficult to understand. In a Swiss cohort study, these correlated factors were obtained at individual level. Controlling for socioeconomic status, women in high skill occupations and of high socioprofessional level were both at increased risk of breast cancer and at increased chance of an early diagnosis. This finding suggests that socioeconomic status and occupation both contribute to inequalities in breast cancer risk and early detection. Interdisciplinary studies with collection of biological, occupational and behavioural information are needed to further explain the causes of socioprofessional inequalities in risk and subtypes of breast cancer.

**Abstract:**

Socioeconomic differences in breast cancer (BC) incidence are driven by differences in lifestyle, healthcare use and occupational exposure. Women of high socioeconomic status (SES) have a higher risk of BC, which is diagnosed at an earlier stage, than in low SES women. As the respective effects of occupation and SES remain unclear, we examined the relationships between occupation-related variables and BC incidence and stage when considering SES. Female residents of western Switzerland aged 18–65 years in the 1990 or 2000 census, with known occupation, were linked with records of five cancer registries to identify all primary invasive BC diagnosed between 1990 and 2014 in this region. Standardized incidence ratios (SIRs) were computed by occupation using general female population incidence rates, with correction for multiple comparisons. Associations between occupation factors and BC incidence and stage at diagnosis were analysed by negative binomial and multinomial logistic regression models, respectively. The cohort included 381,873 women-years and 8818 malignant BC, with a mean follow-up of 14.7 years. Compared with reference, three occupational groups predominantly associated with a high socioprofessional status had SIRs > 1: legal professionals (SIR = 1.68, 95%CI: 1.27–2.23), social science workers (SIR = 1.29; 95%CI: 1.12–1.49) and some office workers (SIR = 1.14; 95%CI: 1.09–1.20). Conversely, building caretakers and cleaners had a reduced incidence of BC (SIR = 0.69, 95%CI: 0.59–0.81). Gradients in BC risk with skill and socioprofessional levels persisted when accounting for SES. A higher incidence was generally associated with a higher probability of an early-stage BC. Occupation and SES may both contribute to differences in risk and stage at diagnosis of BC.

## 1. Introduction

With 2.3 million newly diagnosed cases each year worldwide, breast cancer (BC) is the most frequent cancer in women, accounting for 25% of malignancies [1]. This situation is observed in most high-income countries, including Switzerland, where BC represents 31% of all new cancer cases and accounts for the highest number of potential life-years lost before age 70 [2]. The geographic variability observed in incidence rates suggests that lifestyle factors play an important role. In the United States, risk factors modifiable at menopause account for about one-third of postmenopausal BC [3].

Potentially modifiable risk factors such as chronic exposure to ionizing radiation, artificial light at night, circadian disruptions or to other chemicals, that are usually of occupational origin, have been pointed out [4]. The putative role of occupational exposure to even low-dose ionizing radiation was evidenced in a large cohort study of about 900,000 Finnish women followed from 1971 to 1995 and confirmed in a US cohort of female radiology technologists [5,6]. Whether menopausal status mediates the risk or the highest risk observed for older (i.e., postmenopausal) women is due to working at times when occupational breast doses were considerably higher needs further elucidation. Contrasted results have also been reported on the possible carcinogenic effect of night shift work on BC, notably among nurses [7,8,9]. A recent meta-analysis concluded that night shift work, including long-term shift work, bear little or no effect on BC incidence [10]. Pooled data from five population-based case-control studies in Australia, Canada, France, Germany and Spain found no association between BC and night work in postmenopausal women but an elevated risk for premenopausal women in current or recent night work compared to those who had stopped night work more than two years ago [11]. In this pooled analysis, the risk of BC increased with both duration and intensity of exposure.

More knowledge about occupational exposures is needed in order to reduce the incidence and alleviate the burden of BC. Moreover, women with higher socioeconomic status (SES)-a variable correlated to the occupation-had significantly higher BC incidence and survival rates than women of lower SES [12,13]. Differences in risk, lifestyle and treatment factors as well as in screening attendance and healthcare use have been advanced to explain these socioeconomic inequalities [13,14,15]. It is however often difficult to understand if the observed effect is due to the exposure or is the consequence of the SES.

Therefore, studies combining information on occupational exposure and SES are of great utility. The primary aim of our study was to examine the relationship between occupation and BC incidence taking into account the women’ SES in western Switzerland. The secondary aim was to assess whether the stage at diagnosis differed with occupation when considering SES.

## 2. Materials and Methods

### 2.1. Study Population and Follow-Up

The study included all females aged 18–65 years who resided in western Switzerland (French-speaking cantons of Fribourg, Geneva, Jura, Neuchâtel, Vaud and Wallis) at the time of the 1990 or 2000 census, with known occupation. In Switzerland, the minimum legal age of employment is 15 and the age of majority is 18. The statutory retirement age is 65 for men and 64 for women. Study participants were identified based on the Swiss National Cohort (SNC), a longitudinal research platform with national population coverage estimated at 98.6% [16]. The SNC included records of the 1990 and 2000 Swiss censuses that were linked to mortality, life birth and emigration records, using a combination of deterministic and probabilistic methods [17]. The follow-up started either on December 4th, 1990 (date of the 1990 census) or on December 5th, 2000 (date of the 2000 census) and lasted until the earliest of the following events: emigration date, 85th birthday, death, BC diagnosis date or end of the study (31 December 2014).

### 2.2. Outcome Definition

As main outcome, we considered the primary malignant BC (C50) diagnosed over the period 1990–2014 based on the third edition of the International Classification of Diseases for Oncology (ICD-O-3). Cases were identified using all five cancer registries of western Switzerland (Geneva, Fribourg, Neuchâtel-Jura, Vaud and Wallis). All registries applied international rules for registration of multiple primary cancers [18]. Breast cancer cases from all registries were centralized and their data harmonized in order to enable their linkage with the SNC data. The linkage was performed by a probabilistic linkage procedure. The detailed TNM tumour stage at diagnosis was classified it into stage I, stage II and stage III–IV together [19]. The case selection was applied over the whole period 1990–2014 for the cantons of Geneva, Neuchâtel, Vaud and Wallis and over 2005–2014 for those of Fribourg and Jura who are operating since 2005.

### 2.3. Independent Variables Considered

To better understand the association between BC and occupational and socioeconomic factors, we focused our analyses on four occupation-related variables. The first variable we used was the International Standard Classification of Occupations, 1988 version (ISCO-88). This multi-tiered classification was available in both censuses with the four-digit occupation codes used by the Swiss Federal Statistical Office (SFSO). For this study, we used the one- and three-digit ISCO-88 codes to aggregate 493 four-digit occupations into 9 and 148 occupational groups, respectively. The second variable was the skill level required for the occupation, which was coded into four levels, as defined by Milner et al. [20], and based on the one-digit ISCO-88 codes. It ranges from occupations that require simple, routine physical or manual tasks at level one (i.e., low skill level) to occupations involving performance of tasks that require complex problem solving and decision making at level four (i.e., high skill level). The third variable was the economic activities/industries coded according to the Statistical Classification of Economic Activities in the European Communities (NACE) in the 1990 census and according to the General Classification of Economic Activities (NOGA-95) in the 2000 census, which is a Swiss adaptation of NACE, 1st revision. We recoded both variables into the 17 main categories of NOGA-95. A detailed description of the coding and transcoding of these variables is available elsewhere [21]. Our last variable was the socioprofessional category, a composite variable of the occupation, the situation in the occupation, the highest completed education and the legal form of the company [22]. Because start and end dates of employment were not available, women with a single occupational information contributed with that information throughout their follow-up period, whereas those with a change between the 1990 and 2000 censuses contributed with the first information up to 2000 and with the second thereafter.

### 2.4. Potential Confounders

To account for the substantial variations in the incidence rate of BC over time and across age groups [2] and in mammography screening intensity across Swiss cantons [23], we adjusted our models for age group, calendar period and canton of residence. Given that cancer registries are organised at the cantonal level, adjustment for the latter variable also permitted controlling for potential differences between cancer registries. As BC risk associated with some occupational exposures appear to differ with menopausal status [6,11], sensitivity analyses were performed for all models described hereafter separating premenopausal from postmenopausal women, using age 50 as a dichotomic proxy for menopausal status. Other known potential confounders of BC risk and healthcare use considered in our models were nationality and marital status [24,25,26,27].

### 2.5. Statistical Analyses

All our analyses were performed using STATA V.16 (StataCorp, College Station, TX, USA).

#### 2.5.1. Standardized Incidence Ratios (SIRs)

To identify occupations and economic activities where the incidence of BC differs statistically from that of the working age female population of western Switzerland, we computed standardized incidence ratios (SIRs) as the ratios of the observed to the expected numbers of BC cases for every occupational group. All active women workers were considered to be at risk. The expected number of breast cases was calculated by applying the female BC incidence rate by age (5-year groups) and calendar period (5-year groups) we computed for western Switzerland to the number of person-years for the corresponding calendar period and age group for every occupational and economic activity group. We used a Holm–Bonferroni correction to account for multiple comparisons [28]. We also performed sensitivity analyses over the period 2005–2014 by including BC cases from the more recent Jura and Fribourg cancer registries.

#### 2.5.2. Modelling BC Incidence Rates

To assess the effect of occupational factors while accounting for potential confounders and overdispersion, we analysed BC incidence rates using negative binomial regression. For each woman with a known occupation, we computed person-years at risk that we stratified by calendar period (1990–1995, 1995–2000, 2000–2005, 2005–2010 and 2010–2014) and age group (18–50, 50–70 and 70+). For each occupational variable available (i.e., occupation, economic activity, socioprofessional category and skill level required for the occupation), we constructed univariate models to assess the effect of each variable on the BC incidence rate (Model 1). We adjusted each model first for age, calendar time and canton (Model 2), then for marital status (single, married, divorced, widowed) and nationality (Swiss vs. non-Swiss) (Model 3). Sensitivity analyses were performed by applying Model 3 to the BC incidence rate over the period 2005–2014 (Model 4). All results were expressed as relative risks (RR) with respect to a reference category for each variable and the associated confidence interval at 95% (95%CI) and the Wald test.

#### 2.5.3. Modelling Stage at Diagnosis

To assess the association between stage at diagnosis and occupational variables, we applied multinomial logistic regressions with the same independent variables as described in Model 3 above and tumour stage (stage I, stage II and stage III and IV combined) as the dependent variable. We expressed the results for each occupational factor as the marginal predictions of the probabilities, which are the model-predicted probabilities adjusted (marginalized) on all other factors included in the model. As the proportion of BC of unknown stage was larger before 2000 in some registries, we carried out sensitivity analyses by restricting the modelling of BC stage at diagnosis to the 2000–2014 time period.

## 3. Results

Our study included 381,873 person-years for a mean duration of follow-up of 14.69 years (Table 1). Some 8818 invasive primary BC cases were diagnosed over the 25-year period 1990–2014 in the female population of western Switzerland aged 18–85 years. The study population was predominantly younger than 50 years (59%), Swiss (72%), married (53%) and worked in a low-level management or skilled labour job (56%) with a skill level of second lowest rank (49%).

A comparison across 69 occupational groups (3-digits ISCO-88) showed that women working as building caretakers, window or related cleaners had a lower incidence of BC compared with the reference population (SIR = 0.69, 95%CI: 0.59–0.81) (Figure 1). Occupational groups associated with a statistically significantly increased SIR were legal professionals (SIR = 1.68, 95%CI: 1.27–2.23), women working in social science and related professionals (SIR = 1.29; 95%CI: 1.12–1.49) or employed as secretaries and keyboard-operating clerks (SIR = 1.14; 95%CI: 1.09–1.20).

Regarding the economic branch of activity (Figure 2), a statistically significantly elevated SIR was found only for workers in public administration (SIR = 1.23; 95%CI: 1.11–1.36). Although SIRs above 1 were observed in education, real estate, renting, IT activities, research and development and other business services, and a SIR below 1 for females employed in transport and communication, these effects were not statistically significant once correcting for multiple comparisons (Figure 2). Sensitivity analyses with two additional registries conducted over the shorter time period 2005–2014 confirmed overall the patterns and magnitudes of SIRs observed in the main analyses (Appendix A).

Results from multivariate analyses controlling for time period of diagnosis and sociodemographic factors showed an increasing relative risk (RR) of BC with increasing occupational skill level (RR for highest vs. lowest skill level: 1.39, 95%CI: 1.25–1.54) (Table 2). These findings were consistent across occupation-related variables with statistically significantly reduced risks of 15% to 25% in magnitude for women employed in elementary occupations (RR = 0.78, 95%CI: 0.68–0.88), as unskilled workers (RR = 0.77, 95%CI: 0.66–0.89) and in the economic branches of construction (RR = 0.78, 95%CI: 0.64–0.97), hotels and restaurants (RR = 0.85, 95%CI: 0.76–0.97) and transport and communication (RR = 0.75, 95%CI: 0.66–0.86). Overall, results for the time period 1990–2014 were less pronounced with or without partial adjustment (Models 1 and 2 vs. Model 3, Table 2). Further adjustment for an area-based measure of socioeconomic position did not materially affect the results (Appendix A).

Compared to top managers and independent workers, adjusted RR significantly below 1 were observed only for women younger than 50 in socioprofessional groups of low-level managers and skill labourers (RR = 0.68, 95%CI: 0.53–0.89), unskilled workers (RR = 0.59, 95%CI: 0.44–0.78) and those in unclassified paid employment (Appendix A). Relative risks of BC also differed across economic branches with age at diagnosis. Postmenopausal women (aged 50–85 years) who worked in the construction and premenopausal women (aged 18–49 years) employed in trade, repair of motor vehicles and domestic articles, in hotels and restaurants, or transport and communication had a statistically significantly reduced risk of BC compared to those employed in health and social activities (Appendix A).

After controlling for age, calendar time, canton, marital status and nationality, a gradient was observed between the stage of BC and the skill level or socioprofessional category: the higher the required skill level or socioprofessional category, the higher the probability of being diagnosed with an early-stage BC (Figure 3). This trend was less pronounced when occupation or occupational activity branch were considered (Figure 3a,c). The predicted probability of being diagnosed with an advanced BC (stage III or IV) generally lies between 10 and 18%, regardless of the occupation factor considered. This probability was consistently lower than the probability of being diagnosed with a BC of stage II or I, with exceptions for women employed in compulsory social security (predicted probability of advanced BC: 34.0%, 95%CI: 11.8–56.2%), in domestic services (32.5%, 95%CI: 11.4–53.5%) and working in the branch of electricity, gas and water supply (22.1%, 95%CI: 32.4–41.0%) (Figure 3c). When diagnosed with BC, the highest probability was to have a stage I cancer (34% to 56% across occupational groups). This probability exceeded 50% for professionals (51.3%, 95%CI: 48.2–54.5%), top managers and independent workers (53.1%, 95%CI: 46.0–60.2%), women employed in electricity, gas and water supply (56.6%, 95%CI: 34.6–78.6%), defence (56.0%, 95%CI: 43.1–69.0%), education (51.7%, 95%CI: 48.2–55.1%) and business services such as real estate, renting, IT and R&D activities (50.1%, 95%CI: 46.4–53.9%). Sensitivity analyses conducted over the time period 2000–2014, when completeness of stage of BC was higher, confirmed the results observed for the whole 1990–2014 time period (Appendix A).

Subanalyses for women aged below 50 vs. aged 50+ corroborated the higher probability of being diagnosed with an early-stage BC with higher professional skill level or socioprofessional category (Appendix A). However, the highest predictive probability was associated with a BC diagnosed at stage I for women aged 50+ and at stage II for women younger than 50, this result being more marked by socioprofessional category than by required skill level.

## 4. Discussion

Compared to the general female population, we found an increased risk of BC in three occupational groups that predominantly required highly skilled women and are usually associated with a high socioprofessional level. We also reported a reduced BC risk for women employed as building caretakers, window or related cleaners. Gradients in BC risk with skill and socioprofessional levels largely persisted after accounting for SES. The elevated incidence of BC was associated with a higher probability of having an early-stage tumour for female professionals in top management positions, self-employed or employed in the domain of defence, education, R&D, IT and other business services. Inequalities in BC risk were apparent but less clear when occupational exposure was captured through the economic activity branch (NOGA-95), a proxy which may discriminate less specific occupational exposures and SES.

Occupations and activity sectors associated in our series with increased BC incidence mostly corroborated earlier studies. In France, a case-control population-based study found an elevated odds ratio in white-collar occupations such as managers of wholesale and retail trade as well as for women working in the manufacture of chemicals and other non-metallic mineral products such as ceramics, cement or stone products [29]. French female agricultural workers were at decreased risk of BC with a statistically significant increasing trend with duration of employment [29]. No statistically significant effect was observed in this Swiss study for women employed in the agricultural and fishery branch albeit the incidence of BC was consistently the lowest of all occupations for this sector and for workers in elementary occupations. In contrast, a case-control study in Morocco found a three-fold higher odds ratio of BC for women employed as crop farm laborers and fishery workers, with a positive trend for duration of employment [30]. Because workers in this branch are often considered of poor SES, the SES effect in France might conceal the potential exposure effect observed in Morocco [5,6].

The role of chemical exposure in the BC multifactorial aetiology is not yet fully elucidated but the association could act through alteration of mammary gland development or hormone responsiveness, hormonal tumour promotion or genotoxic action [31]. In a large prospective cohort study of 47,640 US and Puerto Rican women, those with a cumulative exposure to gasoline or petroleum products in the highest quartile cut-off had a doubling in risk of invasive BC compared with women in the lowest quartile group [32]. Swedish female workers exposed to organic solvents (in occupations such as dry cleaners, painters and laboratory technicians) and oil mist (in textile work from spinners’ oil in spinning machines and dyeing processes) were found to be at increased risk of postmenopausal BC, with a risk positively associated with duration of exposure but not with exposure intensity [33]. A Danish study showed a modestly elevated risk of oestrogen receptor negative BC before the age of 50 among women exposed to diesel exhaust [34]. In our study, we observed overall an elevated BC risk for women working to repair motor vehicles and domestic articles but not specifically below age 50, [7,8,9,10,11].

Our observation of an increasing incidence of BC among Swiss women with increasing occupational skill level and socioprofessional category concurred with the results of a recent European meta-analysis [13]. Interestingly, we found that this risk seems independent of SES. The increased incidence of BC persisted, albeit of lesser magnitude, after controlling for SES (highest vs. lowest occupational skill level: relative risks of 39% and 20% without and with adjustment for SES, after controlling for other factors). Whether SES could mitigate the effect of occupational exposures that influences the incidence of BC needs confirmation. Including both occupation-related factors and SES appears thus important in future studies investigating the influence of either SES or occupation on BC risk.

BC was found to be more common and more often diagnosed at an early stage in highly skilled professional women than in workers whose occupation required a low skill level. These effects remained significant after controlling for age, calendar time, canton of residence, marital status and nationality, all factors potentially influencing healthcare use. In Switzerland, screening practices however contributed to the earlier BC diagnosis in highly educated women [35], although differences in use of mammography screening according to SES have strongly attenuated over time [36]. Screening prevalence has recently become higher in unemployed than in employed Swiss women but was 20% lower for independent/artisan workers compared to superior/intermediate professions among women unexposed to organized BC screening programmes, whose participation is virtually free-of-charge [36]. Our estimated highest predictive probability of stage II BC at diagnosis for women younger than 50 years and of stage I for women aged 50 and over concurred with both screening recommendations (from age 50) and the tendency to diagnose more aggressive BC in younger women. Analyses of in situ BC in these Swiss registries may shed further light on the role of screening and healthcare use in general on the observed inequalities in BC risk across occupational and socioprofessional groups.

Mechanisms purported to explain the association of SES with BC aetiology, particularly BC subtypes, include reproductive and environmental factors and chronic stress [37,38]. An increasing number of women, particularly high SES women, who entered the workforce over the last decades, have delayed childbearing, lower parity and higher use of hormonal contraceptives compared to women of lower SES [38]. The increased prevalence of these risk factors mainly among women of greater educational attainment contribute to their higher incidence, particularly of tumours with positive hormone receptors, the more common and less aggressive subtype of BC. Environmental chemicals have been linked to BC risk and low SES women tend to have higher exposure to air pollutants and hazardous jobs [31,39]. Chronic stress may suppress oestrogen production, which could increase the risk of aggressive BC subtypes, and lead to obesity through unhealthy diet and reductions in physical activity, which is an established risk factor for BC in premenopausal women [40]. Low SES women tend also to be more often exposed to chronic stress due notably to financial insecurity, lack of safety or discrimination [40].

In Norway, a study of over one million young women not yet invited for screening reported an increased BC incidence rate among high SES women for both small, localized cancers and tumours with regional spread [41]. The authors’ conclusion pointed toward a real difference in incidence of BC across SES, and not an artifact due to greater opportunistic screening use or higher awareness of early symptoms of BC among high SES women.

Several strengths and limitations in our study can be pointed out. As a population-based cohort, it covered all women diagnosed with invasive BC over a 25-year period and followed-up for a mean of 15 years, irrespective of their occupational exposure. Over the study period, completeness of case ascertainment was high in Swiss cancer registries and BC tumour stage recorded systematically and rigorously according to international rules [42]. Occupation-related data and SES were systematically collected at the individual level by official national censuses and were consistently coded [43]. The performant linkage between these independent data sources enable the concomitant investigation of the relationship between SES, occupation and BC stage. Our results were corroborated by several sensitivity analyses, which suggests that the likelihood of bias in our findings appears low. Finally, the use of four complementary proxies to capture the multidimensional aspects of SES and of occupational categorisation make our results less dependent on the choice of an arbitrary definition of SES.

The main limitation of our study lies in its descriptive rather than causal nature. This means that limited data on covariates were available. No information was collected on BC biology (i.e., in situ BC, hormonal receptors, menopausal status, etc.) and treatment (e.g., hormone replacement therapy, etc.), risk factors or screening behaviour. Data on duration of employments or work exposures to potential health hazards were not available (for instance, exposure to night shift). These unmeasured factors might explain, at least partly, the reduced incidence of BC we observed in caretaking and cleaning occupations. Another limitation is the study exclusion of a substantial proportion of women who were unemployed or for which no known or categorized occupation was available. Although complementary analyses showed no difference in sociodemographic characteristics between included and excluded women, we cannot rule out a potential selection bias as the proportion of BC excluded from the study slightly exceeded the proportion of women excluded [43].

## 5. Conclusions

Results of this first Swiss study on socioprofessional inequalities in BC risk support overall the current but limited evidence that both SES and occupation, when measured at an individual level, contribute to differences in risk and stage of BC. Differences in lifestyle, healthcare use, treatments and occupational exposure are the main explanations for these inequalities by level of SES. Mechanisms to better understand the association between SES and different subtypes of BC, particularly the main aggressive ones, need dedicated interdisciplinary studies with an integrative approach encompassing biological, occupational and health behavioural measurements. In addition, targets of future studies might consider characterizing occupations in terms of exposure to risk factors at large (UV for outdoor workers, sedentary behaviour for office workers, etc.) in order to identify occupational clusters of etiologic risk factors for prevention.

## Figures and Tables

**Figure 1 cancers-14-03713-f001:**
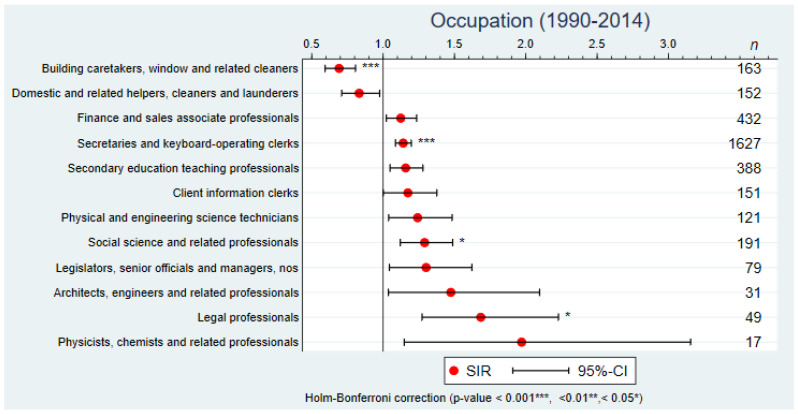
Standardized incidence ratios (SIR) of breast cancer by occupation (3-digits ISCO-88) in western Switzerland, 1990–2014 (only occupations with a statistically significant SIR before the Holm–Bonferroni correction are shown).

**Figure 2 cancers-14-03713-f002:**
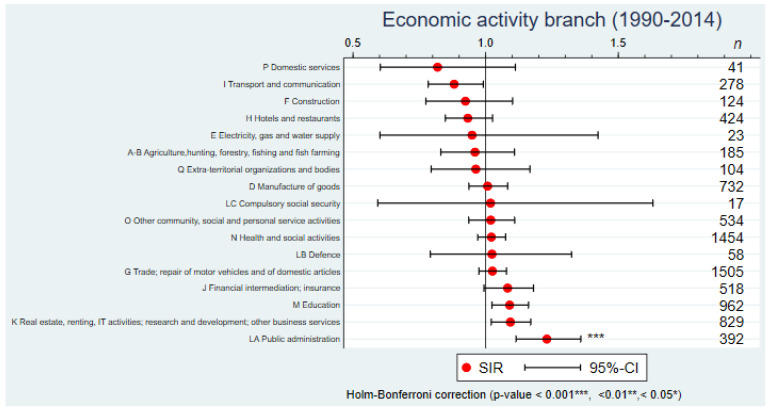
Standardized incidence ratio (SIR) of breast cancer by economic activity branch (17 main NOGA-95 categories) in western Switzerland, 1990–2014.

**Figure 3 cancers-14-03713-f003:**
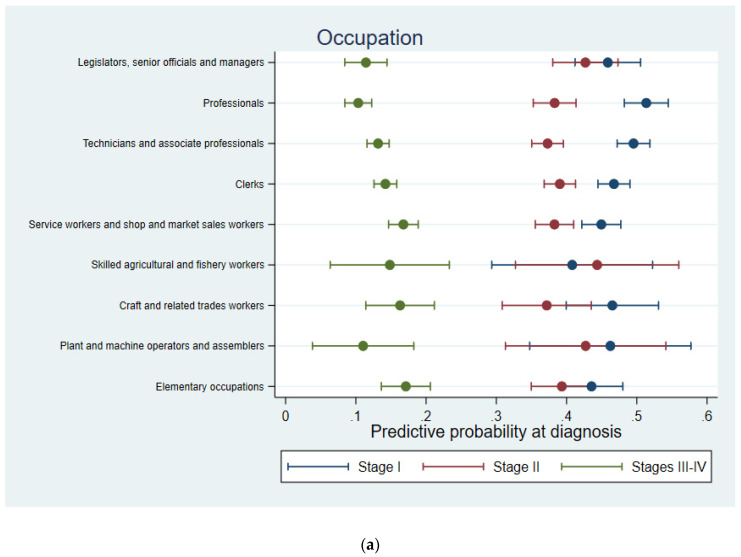
Predictive probability of being diagnosed with breast cancer of stage I, II or III–IV by (**a**) occupation, (**b**) skill level required for the occupation, (**c**) economic activity branch and (**d**) socioprofessional category adjusted for age, calendar time, canton, marital status and nationality for females in western Switzerland, 1990–2014.

**Table 1 cancers-14-03713-t001:** Characteristics of the study population and number of malignant breast cancer cases in western Switzerland, 1990–2014 *.

Characteristics	*n ***	*(%)*	*n* of Breast Cancers	*(%)*
*Total*	381,873	*(100)*	8818	*(100)*
Time at risk (in 100,000 person-years)	56.08			
** *Period* **				
1990–1994	274,696	*(23)*	1409	*(16)*
1995–1999	263,795	*(22)*	2023	*(23)*
2000–2004	224,828	*(19)*	1600	*(18)*
2005–2009	219,306	*(18)*	2042	*(23)*
2010–2014	214,185	*(18)*	1744	*(20)*
** *Age group* **				
Below 50	322,654	*(59)*	2843	*(32)*
Between 50 and 70	203,688	*(37)*	5658	*(64)*
Over 70	24,643	*(4)*	317	*(4)*
** *Nationality* **				
Swiss	279,425	*(72)*	7120	*(81)*
Non-Swiss	106,152	*(28)*	1698	*(19)*
** *Canton of residence* **				
Geneva	113,083	*(29)*	2883	*(33)*
Neuchâtel	44,393	*(11)*	954	*(11)*
Vaud	169,979	*(44)*	3827	*(43)*
Wallis	59,868	*(15)*	1154	*(13)*
** *Socioprofessional category* **				
Top management and independent professions	7450	*(2)*	232	*(3)*
Other self-employed	27,485	*(7)*	823	*(9)*
Professionals and senior management	35,542	*(9)*	926	*(11)*
Supervisors/low level management and skilled labour	234,051	*(56)*	5003	*(57)*
Unskilled employees and workers	100,259	*(24)*	1711	*(19)*
In paid employment, not classified elsewhere	12,689	*(3)*	123	*(1)*
** *Skill level required for the occupation* **				
Lowest skill level	54,831	*(13)*	707	*(8)*
Second lowest skill level	206,451	*(49)*	4281	*(49)*
Second highest skill level	97,576	*(23)*	2124	*(24)*
Highest skill level	65,869	*(16)*	1706	*(19)*
** *Marital status* **				
Single	137,586	*(33)*	1679	*(19)*
Married	216,093	*(53)*	5368	*(61)*
Widowed	11,011	*(3)*	388	*(4)*
Divorced	46,686	*(11)*	1383	*(16)*
Age at the start of follow-up (mean)	36.68			
Age at the end of follow-up (mean)	51.36			
Follow-up duration in years (mean)	14.69			

* 361,105 person-years and 11,179 breast cancers excluded in women unemployed or with no known or categorized occupation. ** The total number of participants was 381,873. As each participant could contribute to several categories of a given variable during the follow-up period, the total of n for each variable is greater than 381,873.

**Table 2 cancers-14-03713-t002:** Relative risk (RR) with confidence interval (95%CI) of breast cancer by occupational, socio-professional, economic activity and skill level category, among females aged 18–85 years in Swiss cantons of Neuchâtel, Geneva, Vaud and Wallis, 1990–2014.

Occupational Variables	Nb Cases	Person-Years (in 100,000)	Models 1 * RR [95%CI]		Models 2 ** RR [95%CI]		Models 3 *** RR [95%CI]	
** *Occupation ^a^* **			***p* < 0.001**		***p* < 0.001**		***p* < 0.001**	
1. Legislators, senior officials and managers	524	2.78	1.00	Ref.	1.00	Ref.	1.00	Ref.
2. Professionals	1182	6.09	1.04	[0.93,1.17]	**1.13**	**[1.00,1.26]**	**1.14**	**[1.01,1.27]**
3. Technicians and associate professionals	2124	13.37	0.90	[0.81,1.00]	1.00	[0.90,1.11]	1.01	[0.91,1.12]
4. Clerks	2283	14.18	0.98	[0.88,1.09]	1.06	[0.95,1.18]	1.06	[0.95,1.18]
5. Service workers and shop and market sales workers	1529	10.83	0.82	[0.74,0.92]	0.92	[0.82,1.03]	0.93	[0.83,1.03]
6. Skilled agricultural and fishery workers	101	0.71	**0.75**	**[0.59,0.96]**	0.82	[0.64,1.05]	0.82	[0.64,1.05]
7. Craft and related trades workers	275	1.71	**0.84**	**[0.71,0.99]**	0.90	[0.77,1.06]	0.92	[0.78,1.08]
8. Plant and machine operators and assemblers	93	0.60	0.80	[0.63,1.01]	0.82	[0.64,1.04]	0.83	[0.65,1.05]
9. Elementary occupations	707	5.80	**0.80**	**[0.71,0.91]**	**0.77**	**[0.68,0.88]**	**0.78**	**[0.68,0.88]**
** *Socioprofessional category* **			***p* < 0.001**		***p* < 0.001**		***p* < 0.001**	
Top management and independent professions	232	0.98	1.00	Ref.	1.00	Ref.	1.00	Ref.
Other self-employed	823	3.68	1.01	[0.86,1.18]	0.97	[0.82,1.14]	0.96	[0.82,1.12]
Professionals and senior management	926	4.83	**0.84**	**[0.71,0.98]**	0.96	[0.82,1.12]	0.97	[0.83,1.14]
Supervisors/low level management and skilled labour	5003	33.31	**0.73**	**[0.63,0.84]**	0.87	[0.75,1.01]	0.89	[0.77,1.02]
Unskilled employees and workers	1711	12.02	**0.71**	**[0.61,0.82]**	**0.75**	**[0.64,0.87]**	**0.77**	**[0.66,0.89]**
In paid employment, not classified elsewhere	123	1.25	**0.67**	**[0.52,0.87]**	**0.67**	**[0.52,0.87]**	**0.70**	**[0.54,0.90]**
** *Skill level required for the occupation* **			***p* < 0.001**		***p* < 0.001**		***p* < 0.001**	
Lowest skill level	707	5.80	1.00	Ref.	1.00	Ref.	1.00	Ref.
2nd lowest skill level	4281	28.03	**1.12**	**[1.02,1.23]**	**1.27**	**[1.15,1.39]**	**1.26**	**[1.15,1.39]**
2nd highest skill level	2124	13.37	**1.12**	**[1.01,1.24]**	**1.30**	**[1.17,1.43]**	**1.29**	**[1.16,1.43]**
Highest skill level	1706	8.88	**1.28**	**[1.16,1.42]**	**1.40**	**[1.27,1.56]**	**1.39**	**[1.25,1.54]**
** *Economic activity branch ^b^* **			***p* < 0.001**		***p* < 0.001**		***p* < 0.001**	
Unknown	636	3.35	1.09	[0.98,1.21]	1.05	[0.94,1.16]	1.06	[0.95,1.17]
A–B Agriculture, hunting, forestry, fishing and fish farming	185	1.03	1.00	[0.84,1.19]	0.97	[0.81,1.15]	0.94	[0.79,1.12]
C Mining and quarrying	2	0.01	0.71	[0.16,3.07]	0.76	[0.18,3.28]	0.76	[0.18,3.28]
D Manufacture of goods	732	4.99	0.90	[0.81,1.00]	0.94	[0.85,1.04]	0.95	[0.85,1.05]
E Electricity, gas and water supply	23	0.16	0.77	[0.49,1.19]	0.82	[0.53,1.28]	0.82	[0.53,1.28]
F Construction	124	0.84	**0.79**	**[0.64,0.97]**	**0.79**	**[0.64,0.97]**	**0.78**	**[0.64,0.97]**
G Trade; repair of motor vehicles and of domestic articles	1505	9.84	0.93	[0.85,1.01]	0.95	[0.87,1.03]	0.95	[0.88,1.04]
H Hotels and restaurants	424	3.37	**0.78**	**[0.69,0.88]**	**0.84**	**[0.74,0.94]**	**0.85**	**[0.76,0.97]**
I Transport and communication	278	2.39	**0.68**	**[0.59,0.79]**	**0.74**	**[0.65,0.86]**	**0.75**	**[0.65,0.86]**
J Financial intermediation; insurance	518	3.69	0.92	[0.82,1.03]	0.97	[0.86,1.09]	0.98	[0.88,1.10]
K Real estate, renting, IT activities; research and development; other business services	829	5.39	0.91	[0.83,1.00]	0.96	[0.87,1.06]	0.98	[0.89,1.08]
LA Public administration	392	1.95	**1.19**	**[1.04,1.35]**	1.11	[0.98,1.27]	1.09	[0.96,1.24]
LB Defence	58	0.32	0.92	[0.70,1.22]	0.99	[0.75,1.30]	0.97	[0.74,1.29]
LC Compulsory social security	17	0.09	0.91	[0.55,1.51]	0.94	[0.57,1.56]	0.94	[0.57,1.55]
M Education	962	5.25	**1.12**	**[1.02,1.23]**	1.05	[0.95,1.15]	1.04	[0.94,1.14]
N Health and social activities	1454	9.07	1.00	Ref.	1.00	Ref.	1.00	Ref.
O Other community, social and personal service activities	534	3.40	0.90	[0.81,1.01]	0.90	[0.80,1.00]	0.91	[0.81,1.01]
P Domestic services	41	0.37	0.84	[0.57,1.23]	0.71	[0.49,1.03]	0.73	[0.50,1.06]
Q Extra-territorial organizations and bodies	104	0.57	1.22	[0.97,1.53]	0.89	[0.70,1.11]	0.94	[0.75,1.18]

Ref.: reference category. ^a^ Occupation is coded on 1 digit using the International Classification of Occupations, version 1988 (ISCO-88). ^b^ Economic activity/industry is coded using the General Classification of Economic Activities (NOGA), based on ISCI third and NACE first revisions. * Univariate model. ** Adjusted for age, period and canton. *** Adjusted for age, period, canton, marital status, marital status x age and nationality. Statistically significant estimates and *p*-values < 0.05 are shown in bold.

## Data Availability

Due to the nature of this research, study participants could not agree for their data to be shared publicly. Supporting data is therefore not available in accordance with ethical and legal requirements.

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
