# Peer review of "Occupational Factors and Socioeconomic Differences in Breast Cancer Risk and Stage at Diagnosis in Swiss Working Women"

_cancers, 2022, doi:10.3390/cancers14153713_

Round 1

Reviewer 1 Report

General comments

This study included data mostly measured at the individual level in a very large cohort of Swiss women over a substantial follow-up period. The Swiss National Cohort includes nearly all Swiss residents and it was linked with Cancer Registries in 5 cantons where 3 cantons had coverage from 1990-2014 and the other two from 2005-2014. Standardized and internationally recognized methods were used for diagnoses and occupations, which provides important common metrics to be able to compare with other study findings. The statistical analyses were generally well done, especially by adjusting for the multiple comparisons and conducting sensitivity analyses.

Major Concerns

1.       My greatest issue is the lack of a definition of socieoeconomic status. It usually includes a combination of income, education, and occupation. However, occupation is one of the variables being compared with SES so the article title and introduction do not seem to be accurate in terms of what is being evaluated. It needs to be clearly stated and that it is not measured at the individual level.

2.      The independent variables defined in Section 2.3 only include 4 occupation variables. The derived variable Socioeconomic position (SEP) is listed as a confounder and includes occupation, income, education and housing conditions measured at the neighborhood level. Thus, there seems to be substantial overlap in terms of adjustment at the area level for occupation too.  Why was occupation included in the derived SEP variable?

3.       In the Simple Summary, the authors state that SES and occupation data are obtained at the individual level, but the SEP is an area-level variable. This is inconsistent with the Socioeconomic position variable as described so the Simple Summary and other statements in the manuscript need to be revised.

4.       The introduction does not include previous literature on SES or occupational factors associated with stage at diagnosis; see for example Wang, Fahui, Lan Luo, and Sara McLafferty. "Healthcare access, socioeconomic factors and late-stage cancer diagnosis: an exploratory spatial analysis and public policy implication." International journal of public policy 5.2-3 (2010): 237

and

Liu, Yang, et al. "Influence of occupation and education level on breast cancer stage at diagnosis, and treatment options in China: A nationwide, multicenter 10-year epidemiological study." Medicine 96.15 (2017).  

5.       The abstract conclusion suggests SES and occupation measures are independent factors  although SES and occupation are correlated based on how they were operationally defined in this study. No interaction terms or stratified analyses were carried out that would confirm this conclusion based on statistical measures.  

6.       No details are provided on how many women were excluded due to missing occupation information, etc. in their administrative data bases. The reasons might not be available but the counts should be reported so the study population can be placed in context with the target population of all eligible women in these cantons during this time period.

7.       How did the results from the 4 different occupation definitions provide insights into why some occupation categories had higher or lower risk?

8.       Why is a detailed work history the most important data to collect to understand the differences based on occupations? Why are not reproductive and lifestyle factors and breast cancer screening and other known etiologic risk factors likely clustered within certain occupations? Being able to characterize the occupations in terms of risk factors might be more appropriate to target in future studies.

Occupations include different exposures such as carcinogenic compounds, UV for outside workers, sedentary behaviours for office workers, physical activity … so occupational labels not as useful perhaps unless these other risk factors that can be quantified too? Why is occupation then so important as it is comprised of multiple risk factors not separately measured?

The discussion section raises a number of factors that could be moved to the Introduction or some that seemed to be not be that relevant to this study. For example, chemical exposure or night shifts as they are not explicitly associated with any of the occupations in this study. 

Minor comments

1.       Table 1 has apostrophes in place of commas and Table 2 seems to have poorly aligned columns and data that make it difficult to understand.

2.       Civil status is not defined and marital status is a more common term.  

3.       Could there be any targeted recommendations to any occupational group based on these findings?

4.       Why were not age-specific breast cancer incidence rates used to calculate the SIRs?

5.       What proportion of women changed occupational categories at the two time points when they reported information at both times? This can provide a sense of the reliability of applying an occupational definition over 10+ years. 

Author Response

We would like to sincerely thank this reviewer for the quality of his/her comments. We feel that his/her suggestions have added merit to the manuscript and significantly contributed to improve it.

Reviewer 2 Report

Dear Authors, Thank you for the opportunity to review this manuscript in exponent of the sample number, I can only suggest some precautions

2.5 I would enrich the section with the description of the statistical methods

Pretentious, remove please: “The typical participant was younger than 50 years (59%), Swiss (72%), married (53%) and worked in a low level management or skilled labour job (56%) with a skill level of second lowest rank (49%) “

Table2: Remove Ref. and use 95%CI

Figure 3, explain the method in the statistical analysis section and below the figure

Reviewer 3 Report

The study aim is interesting and important.

Please consider the following changes:

1. The Introduction section should be more extensive. Please provide background information on cancer incidence; cancer burden; cancer control in Switzerland - that information will be valuable for international readers. 

2. Please explain why this study was limited to " western Switzerland"

3. Please discuss whether the coding system (C50) may have an impact on the results (underdiagnosis or misdiagnosis phenomenon that is observed in some European countries)

4. Please clearly define the limitations section.

5. Please consider adding 2-3 sentences on practical implications of this study.

Round 2

Reviewer 3 Report

The authors applied revisions in line with the comments from the reviewer.